# Effect-Directed Profiling of 17 Different Fortified Plant Extracts by High-Performance Thin-Layer Chromatography Combined with Six Planar Assays and High-Resolution Mass Spectrometry

**DOI:** 10.3390/molecules26051468

**Published:** 2021-03-08

**Authors:** Gertrud E. Morlock, Julia Heil, Valérie Bardot, Loïc Lenoir, César Cotte, Michel Dubourdeaux

**Affiliations:** 1TransMIT Center for Effect-Directed Analysis, and Chair of Food Science, Institute of Nutritional Science, Justus Liebig University Giessen, Heinrich-Buff-Ring 26–32, 35392 Giessen, Germany; Julia.Heil@ernaehrung.uni-giessen.de; 2PiLeJe Industrie, Naturopôle Nutrition Santé, Les Tiolans, 03800 Saint-Bonnet-de-Rochefort, France; v.bardot@pileje.com (V.B.); l.lenoir@pileje.com (L.L.); c.cotte@pileje-industrie.com (C.C.); m.dubourdeaux@pileje.com (M.D.)

**Keywords:** enzyme inhibition assay, antibacterial assay, effect-directed analysis, bioassay, botanicals, health food

## Abstract

An effect-directed profiling method was developed to investigate 17 different fortified plant extracts for potential benefits. Six planar effect-directed assays were piezoelectrically sprayed on the samples separated side-by-side by high-performance thin-layer chromatography. Multipotent compounds with antibacterial, α-glucosidase, β-glucosidase, AChE, tyrosinase and/or β-glucuronidase-inhibiting effects were detected in most fortified plant extracts. A comparatively high level of antimicrobial activity was observed for *Eleutherococcus*, hops, grape pomace, passiflora, rosemary and *Eschscholzia*. Except in red vine, black radish and horse tail, strong enzyme inhibiting compounds were also detected. Most plants with anti-α-glucosidase activity also inhibited β-glucosidase. Green tea, lemon balm and rosemary were identified as multipotent plants. Their multipotent compound zones were characterized by high-resolution mass spectrometry to be catechins, rosmarinic acid, chlorogenic acid and gallic acid. The results pointed to antibacterial and enzymatic effects that were not yet known for plants such as *Eleutherococcus* and for compounds such as cynaratriol and caffeine. The nontarget effect-directed profiling with multi-imaging is of high benefit for routine inspections, as it provides comprehensive information on the quality and safety of the plant extracts with respect to the global production chain. In this study, it not only confirmed what was expected, but also identified multipotent plants and compounds, and revealed new bioactivity effects.

## 1. Introduction

The complexity of the composition of plants and plant extracts gives them properties that are different from and even superior to those of isolated compounds [1,2]. However, in terms of extracts, each extraction process is selective. Distinct secondary metabolites are more or less extracted depending on the conditions used [3,4]. This means that the phytochemical composition of the extracts differs from the plant totum, defined as the entire set of compounds contained in the plant part (used for extraction). Especially, the bioactive metabolites on which the effects in humans and animals depend can highly vary in quantity or be lost or degraded during the production of plant extracts [5,6]. It depends on the quality of the raw material, the applied preprocessing steps, the extraction method used and further treatments during production. Therefore, it becomes clear that extracts of the same plant currently on the market, have not the same quality and composition [6,7]. Information generally given on the production of the so-called botanicals on the market is sparse; almost all specifications lack crucial and more detailed information on the production and final composition of such commercially available plant extracts. This applies in particular to the most important bioactive ingredient profile, which has not yet been harmonized and matched to all relevant active ingredients.

The ipowder^®^ technology was developed to extract as many compounds as possible from the original plant and to increase their content in the final product, solely made of the plant (no further additions such as carrier or filling agents, etc.) [8]. The essential steps of this patented process include contact between the plant material and a solvent in at least one extraction step, and the spray-drying of the extract rich in active compounds on a new batch of the same plant material. The resulting fortified plant material is crushed to form the ipowder^®^, used in dietary supplements and foods with added value. Benefits have been shown in terms of the enrichment of active compounds [5]. While the complex phytochemical composition of such plant extracts can be interesting with regard to biological and therapeutic effects, it makes their analysis and quality assessment extremely challenging. Currently, plant extracts are characterized by identifying and quantifying the well-known characteristic markers (usually one or two) of the selected plant species, whereas in such complex extracts other compounds may also contribute to the effects and overall product quality. Hence, high-performance thin-layer chromatography (HPTLC) hyphenated with planar effect-directed assays (EDA) was found to be a powerful tool for the rapid screening and assessment of such plant extracts in relation to the total bioactivity profile. Two of its many advantages are that it allows the analysis of many plant extracts in parallel in the same chromatographic run and imaging of the different types of the activity responses of the individual compounds without requiring their tedious individual isolation from the plant totum [6,9,10].

In this study, a generic effect-directed profiling was developed for the evaluation and comparison of potential beneficial effects of 17 different ipowder^®^ extracts. HPTLC was combined with five different enzyme inhibition assays and an antibacterial bioassay. The instant bioluminescence of the Gram–negative *Aliivibrio fischeri* bacteria is widely used for the detection of bioactive or antibacterial compounds in environmental samples. The inhibition assays against α-glucosidase, β-glucosidase, acetylcholinesterase, β-glucuronidase and tyrosinase have proven their worth in the search for active plant extracts and secondary metabolites against widespread diseases, such as diabetes, Alzheimer’s and Parkinson’s diseases [6,11,12]. It was hypothesized that these fortified extracts would show pronounced effects in the six selected assays, new information would be obtained by the effect-directed profiles and such bioanalytical profiling would complement the actual analytical tools for product quality control. The important multipotent zones detected were characterized further by the straightforward elution to high-resolution mass spectrometry (HRMS).

## 2. Results and Discussion

### 2.1. HPTLC-UV/Vis/FLD Method Development

A generic nontarget effect-directed profiling was developed for the analysis of 18 plant extracts (IDs **1**–**18**, Appendix A). Analytical methanolic extracts were prepared, as methanol was found to be a good compromise in polarity, based on our experience in other effect-directed studies [6]. No particular analytes were selected or targeted at this stage. The nontarget HPTLC method was developed with the aim of spreading the inherent compounds along the developing distance. Among the different mobile phases tested, two complementary mixtures were selected, i.e., ethyl acetate–toluene–formic acid–water, 16:4:3:2 (MP 1, more polar) versus cyclohexane–ethyl acetate–formic acid, 6:3.8:0.2 (MP 2, more apolar). The combination of the two widened the polarity range and enabled more comprehensive profiling. The physicochemical fingerprints of the inherent compounds in each methanolic extract were detected via multi-imaging at white light illumination (Vis), 254 nm (UV) and 366 nm (FLD). The vertical UV/Vis/FLD profiles (band pattern along each sample track) showed clear differences between the methanolic extracts (Figure 1, IDs **1**–**18**). This outcome was considered a good start to continue with the six effect-directed assays for the detection of the antibacterials and the inhibitors of AChE, α-glucosidase, β-glucosidase, β-glucuronidase, and tyrosinase.

### 2.2. Development of the Effect-Directed Profiling

A neutralization of residual acidic traces on the chromatogram by applying a buffer solution was relevant for both mobile phase mixtures containing formic acid. The comparison of the FLD images before versus after the neutralization step (Figure 1C versus Figure 1D) showed that the additional buffer salt load on the plate reduced the fluorescence intensity of most zones. In addition, it altered the fluorescence nuance and intensity of some zones (e.g., IDs **16**–**18** in MP 1). Such fluorescence signal shift can be explained by the pH change (from acidic to neutral) or inorganic traces in the buffer solution. MS grade HPTLC plates have been available on the market for some years. These plates are known to have a lower layer thickness and were therefore investigated for their applicability in the effect-directed profiling. The rule of thumb was that half the volume of (expensive) assay solution was needed as with normal plates. However, the correct volume also depended on the humidity of the day, and thus layer. On humid days, the application of even less assay solution volume was recommended to avoid an excessive liquid layer. The more water was preadsorbed from the ambient air, the less assay volume could be adsorbed and penetrate to the deeper layer. On days with higher humidity (>50%), the plate was therefore dried (e.g., 40 °C, 1 min). The subsequent assay application started with the *A. fischeri* bioassay. This bioassay is frequently used [13], and gives a first overview on sample compounds with influence on the energy metabolism of the nonpathogenic Gram–negative bacteria. Based on our experience, this bioassay often indicates most bioactive compounds present, if compared to other assays. Thus, this instant bioassay is helpful for a cost-efficient selection of the first effect-relevant sample amount to be applied. For applied 200 µg extract per band, antibacterial compound zones were detected. This was a good start for this and all the other assays, although later adjustment or fine-tuning may be required.

#### 2.2.1. Antibacterial Activity

In the Gram–negative *A. fischeri* bioautogram, active compounds can have a positive or negative influence on the bioluminescence of the bacteria, resulting in bright or dark zones. Of each plant extract, 200 µg were applied which allowed a direct comparison of the antibacterial activity between the different samples (Table 1). Many antibacterial bioluminescence-reducing compound zones were observed in 15 out of the 17 fortified plant extracts (Figure 2A, depicted as greyscale image, IDs **1**–**18** in Appendix A, respective UV/Vis images in Appendix A). In particular, the fortified extracts of *Eleutherococcus* (**4**), grape pomace (**15**), passiflora (**16**), artichoke (**17**) and *Eschscholzia* (**18**) had a strong Gram–negative antibacterial activity. Strong antibacterial zones were also detected for the fortified green tea extract (**1**), which were 2–3-fold stronger, when visually compared to the nonfortified green tea extract (**5**). This was in good agreement with the specified drug-to-extract ratio of 2:1 for the production of the fortified green tea extract (Appendix A). An increased content of chemical markers was recently reported for ipowder^®^ extracts [5], which was also confirmed by this comparison (ID **5** versus **1**). In both green tea samples, two dominant antibacterial zones were evident at *hR*_F_ 80 and 89 (the latter marked as zone **V**) in the stronger eluting MP 1 system, which were located at *hR*_F_ 5 and 11 (zone **V**) in the weaker eluting MP 2 system. This illustrates clearly that the polarity range was complementary to each other, and thus selected well for a comprehensive profiling. Comparatively fewer and also less intense antibacterial zones were observed in the fortified extracts of lemon balm (**2**), rosemary (**3**), red vine (**7**), meadowsweet (**9**) and *Echinacea* (**10**). However, additional apolar antibacterial compounds of especially hops (**14**) and also rosemary (**3**), and to some extent for valerian (**8**) and blackcurrant (**11**) were first revealed with the apolar MP 2 system. Dark zones, which eluted almost with the solvent front of the stronger eluting MP 1, were now well separated and spread along the developing distance. In contrast to these positive findings and considering the same sample amount applied, black radish (**12**) and horse tail (**13**) were the only two fortified extracts with no obvious antibacterial activity in both investigated polarity ranges. It is worth noting that an enhancement of the bacterial bioluminescence was observed for more apolar compound zones in the MP 2 bioautogram for yerba mate (**6**) at *hR*_F_ 83 and meadowsweet (**9**) at *hR*_F_ 77 as well as for a few further plants such as lemon balm (**2**) and rosemary (**3**). The bright zones may highlight a prebiotic activity. Recently, some fatty acids have been proven to enhance the bioluminescence of the Gram–negative *A. fischeri* bacteria [7].

While the antibacterial effect is well known for some of these plants, it is much less described for others. Among the plants with the highest activity against *A. fischeri* bacteria, the hop plant is recognized as an excellent source of antibacterial compounds [14,15]. This is confirmed by our study in view of the number and intensity of antibacterial zones observed for hops (**14**) in the MP 2 bioautogram. The high antibacterial activity of numerous compounds in *Eleutherococcus* root (**4**) is shown for the first time. Data on the antibacterial effect of this plant are rare [16,17]. The strong antibacterial effect observed for *Passiflora incarnata* (**16**), is only supported by a reported effect against *Helicobacter pylori* [18]. As for the observed antibacterial activity of *Eschscholzia* (**18**), its alkaloids have demonstrated antifungal [19] and some even antibacterial potential [20,21]. In accordance with our findings is also the well documented antibacterial activity of green tea [22,23] and grape pomace [24]. Extracts and essential oils of lemon balm [25] and rosemary [26] have also proven antibacterial activity, which confirms our results. A few more studies confirm the outcome of our profiling, which have shown antibacterial effects for artichoke [27], meadowsweet [28,29], red vine leaf [30], *Echinacea* species [31], blackcurrant [32], yerba mate [33], and valerian species, whereby the only study found for *Valeriana officinalis* was on the essential oil of its root [34].

#### 2.2.2. Enzyme Inhibiting Activity

In the five different enzyme inhibition assay autograms (Figure 2B–F), enzyme inhibitors appeared as colorless (white or yellowish) zones on a violet, grey or blue background. A pronounced enzyme inhibition was observed for the majority of the 18 tested plant extracts. For the same plant extract, the brightest inhibitory zones were often detected in several enzyme assays, and thus found to be multipotent. In contrast, red vine (**7**), black radish (**12**) and horse tail (**13**) were three plant examples with comparatively very weak enzyme inhibition profiles across all five enzyme assays performed. The comparison of the fortified green tea extract (**1**) versus nonfortified extract (**5**) confirmed that the same enzyme inhibiting zones were present. However, in each case, a stronger effect was evident for the fortified extract across all five inhibition assays. This shows the potential of this bioanalytical profiling for quantitative studies [7]. This stronger effect of the fortified extract observed in this profiling was also consistent with target analysis in that a higher concentration of chemical markers was found in the fortified extracts [5].

In the α-glucosidase inhibition autograms, the bright zones indicated α-glucosidase inhibitors, and thus potential antihyperglycemic components in the 18 tested plant extracts (200 µg each; Figure 2B; Table 1, respective UV/FLD images in Appendix A). Inhibiting compound zones were detected for all plants but black radish (**12**) and horse tail (**13**). Interestingly, most of the plant ingredients with anti-α-glucosidase activity were also able to inhibit β-glucosidase (200 µg/band; Figure 2C). The horizontal pattern recognition (along the so-called substance windows in HPTLC) showed that several intense inhibition zones observed in the α-glucosidase assay were also detected in the β-glucosidase assay, indicating the same main inhibitors. However, a different pattern or less intense β-glucosidase inhibitory zones were detected for example for rosemary (**3**) and hops (**14**) in the MP 2 autograms (Figure 2C versus Figure 2B). Depending on the plant, the selective α- or β-glucosidase inhibitions by compounds were similar, but also different. Many studies have been conducted on the anti-α-glucosidase activity of plant extracts and their compounds, while comparatively less literature is available on the also important anti-β-glucosidase effects [35,36]. In the two glucosidase autograms, the green tea profiles (ID **1** and **5**) showed two very intense (more brownish) active zones. These were identical to the two previously discussed antibacterial zones at *hR*_F_ 80 and 89 (zone **V**, MP 1) and *hR*_F_ 5 and 11 (zone **V**, MP 2). Studies have shown that green tea contains potent α-glucosidase inhibitors [37,38,39]. To the best of our knowledge, there is no data showing an effect of green tea on β-glucosidase in the literature but the present study. Meanwhile, this effect was studied and assigned to the catechins in green tea [6]. Zones at a similar *hR*_F_ value were identified in both glucosidase assays for meadowsweet (**9**) which has known effects on α-glucosidase [40], but none on β-glucosidase. Hence, the anti-β-glucosidase activity of meadowsweet is evident for the first time. An intense inhibiting compound zone at *hR*_F_ 90 (zone **VI**) was observed for lemon balm (**2**) and rosemary (**3**) in both MP 1 autograms. The same intense zone was evident at *hR*_F_ 11 (zone **VI**) in both MP 2 autograms. Two further less intense zones were observed at *hR*_F_ 62 and 67 (MP 1) for lemon balm, and much weaker in the response for rosemary. However, in both MP 2 autograms of rosemary, two further intense α-glucosidase and comparatively weaker β-glucosidase inhibitors were revealed at *hR*_F_ 73 (zone **IX**) and 83 (zone **X**); both eluted with the solvent front in the MP 1 system. The inhibitory effects of lemon balm and rosemary on α-glucosidase have been previously observed [12,41], whereas to the best of our knowledge, their anti-β-glucosidase activity is reported here for the first time. The profiles of yerba mate (**6**) and artichoke (**17**) each show an inhibitory zone at *hR*_F_ 45 (zone **I**, MP 1) in both glucosidase assays. Two further inhibiting zones were revealed at *hR*_F_ 71 (zone **II**) and 85 (zone **IV**) for yerba mate (**6**), and at *hR*_F_ 82 (zone **III**) and 92 (zone **VII**) for artichoke (17) in both MP 1 autograms. While the α-glucosidase inhibitory activity of yerba mate and artichoke has been previously shown [42,43,44], their β-glucosidase inhibition has not been reported so far. Hops (**14**) showed the strongest α-glucosidase inhibition profile across the tested 17 fortified extracts in the MP 2 autogram, and thus, hops were found to be a comparatively strong α-glucosidase inhibitor. A previous study has demonstrated the α-glucosidase inhibitory activity of hops [45]. The comparatively weaker β-glucosidase inhibition has not been reported so far.

AChE inhibitory zones in the applied extracts (200 µg/band; Figure 2D; Table 1) were especially observed for green tea (**1** and **5**), lemon balm (**2**), rosemary (**3**), yerba mate (**6**), meadowsweet (**9**), and *Eschscholzia* (**18**). Meanwhile, the AChE inhibitory effect was studied and assigned to the catechins in green tea [6]. For *Eschscholzia*, three active zones, at *hR*_F_ 52, 71 and 84, were evident in the MP 1 autogram and three further, at *hR*_F_ 10, 26 and 90, in the MP 2 autogram. In *Eschscholzia*, the AChE inhibiting activity was strongest among the five enzyme assays examined. Further weaker AChE inhibiting zones were detected for valerian (**8**), *Echinacea* (**10**), grape pomace (**15**) and artichoke (**17**). Green tea [46], lemon balm [47], rosemary [48], yerba mate [49], meadowsweet [50], artichoke [44] and grape pomace [51] were reported to inhibit AChE, which confirms our profiling results. For *Eschscholzia*, no prominent AChE inhibition was observed for 14 isolated alkaloids [52]; however, using a simple methanol extraction of the plant-fortified extract in our study, an intense AChE inhibiting zone at *hR*_F_ 84 (MP 1) and five weaker zones (MP 1/2) were observed.

For twice the sample amount applied (400 µg/band; Figure 2E; Table 1), the tyrosinase inhibitory activity was considered to be comparatively lower, if compared to the previous assays. Distinct tyrosinase inhibition zones were detected for green tea (**1** and **5**), lemon balm (**2**), rosemary (**3**), yerba mate (**6**), meadowsweet (**9**), grape pomace (**15**) and artichoke (**17**). Antityrosinase activity was reported for green tea [53], lemon balm [54], rosemary essential oil [55], meadowsweet [50] and grape pomace [56]. No proof of the tyrosinase inhibition was found in the literature for yerba mate, artichoke and *Eschscholzia*. In this profiling study on fortified extracts, tyrosinase inhibitors were detected for the first time. For comparatively weak responses (e.g., for *Eschscholzia*), higher sample amounts should be applied in future studies. Minor compounds could be better detected when higher sample amounts were applied on the plate, but the autogram was then overloaded for the most active samples.

For the 1.5-fold extract amount applied (300 µg/band; Figure 2F; Table 1), the β-glucuronidase was inhibited by almost all fortified plant extracts, as evident especially in the stronger eluting MP 1 autogram. The β-glucuronidase inhibition was comparatively more pronounced (the profile was already overloaded by the strong signals) for rosemary (**3**), a bit less intense for lemon balm (**2**) and then green tea (**1**), yerba mate (**6**) and meadowsweet (**9**). The inhibitory activity was more moderate for the other plants, but for *Echinacea* (**10**), black radish (**12**) and horse tail (**13**) an effect was not or hardly observed. In rats, the administration of green tea extract reduced the activity of bacterial β-glucuronidase [57], which confirms our profiling result. Blackcurrant (**11**) was also shown to inhibit this enzyme [58], which also showed a moderate β-glucuronidase inhibition in our study. For the other plants tested, a β-glucuronidase inhibition has not been reported so far. It is first recognized and illustrated by this effect-directed profiling.

### 2.3. Characterization of Active Zones I–XI by HPTLC–HESI–HRMS

Eleven prominent multipotent compound zones detected in the different assays (**I**–**XI**, numbered from polar to apolar) were further characterized by HPTLC–HESI–HRMS. Therefore, six plant extract samples were applied twice (two sets) on the same plate. After plate cut, one plate part was subjected to the bioassay. On the other plate part (Figure 3), the eleven zones were marked according to their bioautogram responses. The high-resolution mass spectra of these zones were recorded and evaluated (Figure 4).

The exact mass signals found were assigned to molecular formulas and tentative candidates in agreement with the literature data (Table 2). In green tea (**1**) and meadowsweet (**9**), the zone **V** at *hR*_F_ 89 (MP 1) and 11 (MP 2) was a coelution of catechins, whereby the most pronounced mass signal (base peak) was obtained at *m*/*z* 289.0720 [M − H]^−^, tentatively assigned as (epi)catechin (C_15_H_14_O_6_). In the MP 1 system, the different catechins in green tea also coeluted in two zones—the zone **V** and the active zone below (*hR*_F_ 80). Catechins in green tea and meadowsweet were therefore in part responsible for the activities of both plants. The antibacterial activity of green tea is ascribed in particular to galloylated catechins such as (−)-epicatechin gallate and (−)-epigallocatechin gallate [6,22,23].

The reported antibacterial activity of meadowsweet has not been attributed to a structural class or particular molecule [28,101,102]. Of note is the fact that meadowsweet flowers, the plant part used to make the fortified extract, were shown to contain catechins [40], which confirms our assignment.

For zone **VIII** at *hR*_F_ 100 (MP 1) and 24 (MP 2) in green tea (**1**) and meadowsweet (**9**), the base peak at *m/z* 169.0142 [M − H]^−^ was assigned to gallic acid (C_7_H_6_O_5_). Opposite to zone **V** (catechins), the gallic acid zone **VIII** showed no antibacterial effect against Gram–negative bacteria. However, both zones (**V**, catechins and **VIII**, gallic acid) were identified as potent inhibitors of α-glucosidase, β-glucosidase, AChE and tyrosinase. The respective β-glucuronidase inhibition was comparatively low for zone **V**. A more brownish instead of bright inhibition zone was revealed for the catechins (zone **V** and zone below). This was caused by the substrate for high catechin amounts, which was proven in detail in another study and then substituted by another substrate [6]. Our results were confirmed by reports on catechins and derivatives to be responsible for the inhibition of α-glucosidase [59] and β-glucosidase [60]. Gallic acid was reported as inhibitor of α-glucosidase [63,64]. The β-glucosidase inhibition was, to the best of our knowledge, first reported here. Catechins and gallic acid were reported to be AChE and tyrosinase inhibitors [53,61,65,66]. The β-glucuronidase inhibition was reported for catechins [62]. Gallic acid has no reported effect on this enzyme, which complies with our result. For zone **VI** at *hR*_F_ 90 (MP 1) and 11 (MP 2) in lemon balm (**2**) and rosemary (**3**), the base peak at *m/z* 359.0775 [M − H]^−^ was assigned to rosmarinic acid (C_18_H_16_O_8_), which was reported as a marker compound in both plants [103]. The rosmarinic acid zone **VI** was clearly detected in all six assays.

Zone **IX** at *hR*_F_ 100 (MP 1) and 73 (MP 2) in rosemary (**3**) showed a mass signal at *m/z* 329.1761 [M − H]^−^ and as base peak its oxidized form at *m/z* 345.1709 [M + O − H]^−^. These mass signals were assigned to carnosol (C_20_H_26_O_4_). The zone **X** at *hR*_F_ 100 (MP 1) and 83 (MP 2) showed a base peak at *m/z* 455.3534 [M − H]^−^ and was assigned to oleanolic/ursolic acid (C_30_H_48_O_3_). The carnosol zone **IX** and oleanolic/ursolic acid zone **X** were very intense in the antibacterial bioassay as well as α-glucosidase and β-glucuronidase inhibition assays, whereas both were much less intense inhibitors in the other assays. Rosmarinic, oleanolic and ursolic acids as well as carnosol have been identified as antibacterial agents [67,68,69,70,71,74,75,78,104], which confirms our results. The inhibitory effects of rosmarinic acid (pure and from rosemary and lemon balm) have been observed on α-glucosidase [12,40,41], which is also in agreement with the intense inhibition observed in our study. Oleanolic/ursolic acid are known α-glucosidase inhibitors [79,80], but in our study, the effect of rosemary is directly related to the presence of these compounds for the first time. In line with our results, carnosol from rosemary was identified as an α-glucosidase inhibitor; its oral administration significantly reduced the postprandial blood glucose levels of normal mice [76]. Rosmarinic, oleanolic and ursolic acids are also known inhibitors of AChE [72,81,82,104] and tyrosinase [73,83,84,104], which was confirmed by our results. Their ability to inhibit β-glucosidase and β-glucuronidase was not known so far. The direct effect of carnosol on AChE had not been shown so far, whereas its effect on tyrosinase has been reported [77].

Zones **I**, **II** and **IV** (*hR*_F_ 45, 71 and 85 in MP 1, respectively; all *hR*_F_ 0 in MP 2) observed in yerba mate (**6**) were assigned to chlorogenic acid (C_16_H_18_O_9_, base peak at *m/z* 353.0881 [M − H]^−^), caffeine (C_8_H_10_N_4_O_2_, base peak at *m/z* 217.0697 [M + Na]^+^) and dicaffeoylquinic acid (C_25_H_24_O_12_, base peak at *m/z* 257.0563 [M − 2H]^2−^), respectively. The chlorogenic acid zone **I** was also observed in artichoke (**17**), in addition to zone **III** (*hR*_F_ 83 in MP 1 and *hR*_F_ 0 in MP 2) assigned to cynaratriol (C_15_H_22_O_5_, base peak at *m/z* 305.1359 [M + Na]^+^) and zone **VII** (*hR*_F_ 92 in MP 1 and *hR*_F_ 24 in MP 2) assigned as 3-*O*-feruloyl quinic acid (C_17_H_20_O_9_, base peak at *m/z* 369.1307 [M + H]^+^). These zones were detected in all five enzyme assays, except for the lack of AChE inhibition of 3-*O*-feruloyl quinic acid (zone **VII**). In previous studies, chlorogenic acid showed antibacterial activity against Gram–negative bacteria [85] and caffeoylquinic acids were in the top 10 identified antibacterial compounds of yerba mate [94], whereas the antibacterial effect of caffeine is first reported here. The previously reported antibacterial effects of artichoke were ascribed to phenolic compounds without further specification [27,105]. Chlorogenic and 3-*O*-feruloyl quinic acids are phenolic compounds, but not cynaratriol, a known sesquiterpene of artichoke, whose antibacterial activity has not yet been reported. Chlorogenic acid and dicaffeoylquinic acid have known anti-α-glucosidase [86,87,95], anti-AChE [88,89,96] and antityrosinase [90,97] activities. No information about a potential effect of caffeine on α-glucosidase, which seems to be new information, was found in the literature. The reported inhibition of AChE [92] and tyrosinase [93] by caffeine was confirmed by our results. The β-glucosidase inhibiting activity for these compounds and the cynaratriol and 3-*O*-feruloyl quinic acid inhibitory effects on the different enzymes are described for the first time. With regard to the β-glucuronidase inhibitory activity, only chlorogenic acid [91] and caffeoylquinic acid [98] have been reported to have an inhibitory effect on this enzyme so far.

Zone **XI** in *Eschscholzia* (**18**) was assigned to coeluting fatty acids, i.e., linoleic acid (C_18_H_32_O_2_, base peak at *m/z* 279.2330 [M − H]^−^) and linolenic acid (C_18_H_30_O_2_, base peak at *m/z* 277.2175 [M − H]^−^). This linoleic/linolenic acid zone was active in the tested assays, which is in accordance with a recent study [7]. Our results were confirmed by the reported antibacterial [99] and α-glucosidase inhibiting effects [100] of linoleic and linolenic acids.

### 2.4. Outlook

It is worth mentioning that the developed effect-directed profiling can also be used for quantitative studies based on the enzymatic or biological response or for the calculation of IC_50_ and MIC values, as recently shown in other studies [106,107,108,109]. Planar assays have proven to be just as reliable as current microtiter plate assays, but additionally offer all the advantages of separating a complex sample [110,111]. HPTLC-EDA workflows are highly efficient in time and costs (by a factor of 7 less costs [111]), robust with regard to matrix (minimalistic sample preparation) and reveal new options, such as the simultaneous detection and differentiation of agonistic and antagonistic effects of individual compounds in a mixture [112].

## 3. Materials and Methods

### 3.1. Chemicals and Reagents

Fast Blue B salt (95%) was purchased from MP Biomedicals, Eschwege, Germany. We obtained 5-Bromo-4-chloro-3-indolyl β-d-glucuronide sodium salt (X-Gluc, ≥98%) from Carbosynth, Compton–Berkshire, UK. α-Glucosidase (from *Saccharomyces cerevisiae*, 1000 U/vial), acarbose (for pharm.), tyrosinase (from mushroom, ≥1000 U/mg, 25 kU/vial), β-glucuronidase (from *Escherichia coli*, 5000 U/vial), imidazole (≥99.5%), acetylcholinesterase (AChE, from *Electrophorus electricus*, ≥245 U/mg solid, 10 kU/vial), 3-[(3-cholamidopropyl)-dimethylammonio]-1-propanesulfonate (CHAPS, ≥98%), di-ammoniumhydrogen phosphate (99%), pepton from casein (tryptone, for microbio.), cyclohexane (HPLC grade), sodium acetate, monopotassium phosphate, magnesium sulfate heptahydrate and sodium chloride (all p. a. quality and waterfree) were obtained from Fluka or Sigma–Aldrich, Steinheim, Germany. *Aliivibrio fischeri* bacteria (no. 7151) were from the German Collection of Microorganisms and Cell Cultures, Düsseldorf, Germany. 2-Naphthyl-α-d-glucopyranoside (95%) was delivered by Fluorochem, Hadfield Derbyshire, UK. Bovine serum albumin (BSA, fraction V, ≥98%), acetic acid (Ph. Eur.), dipotassium hydrogen phosphate (waterfree, ≥99%), sodium dihydrogen phosphate monohydrate (≥98%), glycerol (86%), potassium dihydrogen phosphate (99%), sodium hydroxide (≥98%), disodium hydrogen phosphate (≥99%), polyethylenglycol (PEG) 8000, koji acid (>98%), acetic acid (Ph. Eur.), hydrochloric acid (HCl, 37%, purest), tris(hydroxymethyl)aminomethane (Tris, ≥99.9%) were purchased from Carl Roth, Karlsruhe, Germany. Methanol, ethanol and cyclohexane (all HPLC grade) as well as formic acid (99%, LC–MS) were obtained from vwr, Darmstadt, Germany. Toluene (HPLC grade) was from Promochem, LGC Standards, Wesel, Germany. HPTLC plates silica gel 60 F_254_ MS grade (20 cm × 10 cm, Lot HX69361434) and citric acid monohydrate were obtained from Merck, Darmstadt, Germany. 2-Naphthyl-β-d-glucopyranoside (95%) and β-glucosidase (from almonds, 3040 U/mg) was purchased from ABCR, Karlsruhe, Germany. 1-Naphthyl acetate (≥98%) was delivered by AppliChem, Darmstadt, Germany. d-Saccharolactone (≥98%) and (2S)-2-amino-3-(3,4-dihydroxyphenyl)propanoic acid (l-DOPA, 96%) was from Santa Cruz Biotechnology, Santa Cruz, CA. Ethyl acetate (≥99.8%) and yeast extract powder (Chemsolute, for microbiol.) were purchased from Th. Geyer, Renningen, Germany. Bidistilled water was prepared using the Destamat Bi 18 E system of Heraeus, Hanau, Germany. For incubation, a polypropylene box (27 cm × 16 cm × 10 cm, KIS, ABM, Wolframs–Eschenbach, Germany) was covered by wetted filter papers at the bottom and sides.

### 3.2. Production of Fortified Plant Powders

The harvest of each plant was performed at an optimal period as specified in Appendix A. From the respective field, each plant was collected and either frozen (grape pomace) or dried within 12 h (roots were washed before treatment to remove soil). Thereby, each plant material was specified (Appendix A) and agricultural treatments were documented (data not shown). Dried plants were bagged into paper or mesh bags, frozen grape pomace was stored in plastic containers. Each extract was produced on an industrial scale (0.6–2.4 t per batch), depending on the plant material (PiLeJe Industrie, Saint–Bonnet de Rochefort, France). The same patented ipowder^®^ process, in which a plant extract was added to the same ground plant material [8], was applied to each plant. However, temperature and duration of the extraction varied depending on the plant material (Appendix A). Briefly, a finely crushed dry plant material was extracted using water, except for the frozen grape pomace, which was extracted using ethanol–water (3:7, *v*/*v*; for stability of the extract). The ratio of plant to first native extract was 4–7 to 1 depending on the plant material. Each plant extract was filtered, concentrated and spray-dried over a new batch of the same ground plant material. For the resulting fortified ipowder^®^ plant extracts, the final ratio of dry plant to final extract was 2–5 to 1, depending on the plant material (listed in detail in Appendix A). For evaluation of the fortification success, the green tea raw material was also investigated and compared with the respective ipowder^®^ extract (Appendix A, green tea leaf powder ID 5 used to make the fortified extract ID 1). More than 95% of the plant powder particles were less than 500 µm in size. The powders are stable for 5 years when stored at room temperature in a dry and dark place.

### 3.3. Extraction of the Plant Powders

Each fortified plant powder (1 g each) and the green tea leaf powder were suspended in 10 mL methanol, extracted by ultrasonication for 30 min and centrifuged at 3000× *g* for 15 min. Each supernatant, of which an aliquot was transferred to a sampler vial for analysis, was stored in the dark at −18 °C (stable for 2 years). Note that the analytical extract of green tea leaf powder (the raw material ID 5 used to make the fortified extract ID 1, Appendix A) is referred to as nonfortified extract in the text.

### 3.4. HPTLC–UV/Vis/FLD Method

All methanolic extracts (2 µL/band each, except 3 µL/band for the β-glucuronidase assay and 4 µL/band for the tyrosinase assay) were applied as 8-mm band on the HPTLC plate (distance to lower edge 8 mm, distance to side edge at least 15 mm, dosage speed of 150 nL/s, Automatic TLC Sampler ATS 4, CAMAG, Muttenz, Switzerland). In a twin trough chamber or Automated Development Chamber (ADC 2, CAMAG), the plate was developed using one of the two solvent mixtures (10 mL each) up to a migration distance of 70 mm (measured from the lower plate edge, taking ca. 35 min). Either a more polar solvent mixture MP 1 consisting of ethyl acetate–toluene–formic acid–water 16:4:3:2 (*v*/*v*/*v*/*v*), or a more apolar solvent mixture MP 2 consisting of cyclohexane–ethyl acetate–formic acid 6:3.8:0.2 (*v*/*v*/*v*) was used. The chromatogram was dried in the ADC 2 for 20 min (MP 2) or 30 min (MP 1). Documentation was performed at white light illumination (Vis), UV 254 nm and FLD 366 nm (TLC Visualizer, CAMAG).

### 3.5. Neutralization of Acidic Traces

The formic acid traces which remained on the chromatogram were neutralized. Briefly, 0.75 mL phosphate buffer solution (8 g disodium hydrogen phosphate in 60 mL water, adjusted to pH 7.5–7.8 with 0.1 M citric acid, ad 100 mL) was piezoelectrically sprayed (yellow nozzle, level 6, hood and tray for 20 cm × 10 cm plates, Derivatizer, CAMAG) according to a recently optimized workflow [104]. The plate was dried in a stream of cold air (2 min by hair dryer, then 15 min by ADC 2).

### 3.6. Effect-Directed Profiling

Six chromatograms were prepared in parallel. For each assay, respective positive controls were applied above the solvent front at the upper plate edge of the neutralized, dried chromatogram [6]. For example, three bands of an aqueous saccharolactone solution (100, 150 and 200 ng/band) were applied for the β-glucuronidase assay. Each plate was subjected to the respective assay solutions or suspensions by piezoelectric spraying (placed on a filter paper sheet, if not stated otherwise, yellow nozzle, level 6, Derivatizer, CAMAG). After the first spraying, the bottom side of the nozzle was manually dried with a lint-free tissue to avoid a dropping on the plate during the second spraying. For incubation (at 37 °C, if not stated otherwise), each plate was placed horizontally in a humid poly-propylene box (premoistened for 30 min at room temperature with 35 mL water spread on filter papers aligned on walls and bottom). Drying was performed in a stream of cold air (hair dryer). If not stated otherwise, documentation was performed at white light illumination in the reflectance mode (TLC Visualizer, CAMAG). Aliquoted enzyme and l-DOPA substrate solutions were stored at −18 °C, whereas other solutions were stored in the dark at 4 °C; all were stable for several months.

#### 3.6.1. HPTLC–*A. fischeri* Bioassay

The HPTLC–*A. fischeri* culture was prepared according to DIN EN ISO 11348–1, 2009. After a day, the progress of the bacterial growth (increasing bioluminescence) was visually controlled by shaking the culture flask in a dark room. When the emitted green-blue light was brilliant, the *A. fischeri* suspension (3.5 mL) was piezoelectrically sprayed on the chromatogram. Thereby, the vapor settling down phase was interrupted to take out the plate (hood was closed again to suck out the rest of the vapor into the trap), which was placed wet into the BioLuminizer cabinet CAMAG). The bioautogram was documented over 30 min (exposure time 1.0 min, trigger interval 3.0 min). Dark or bright zones indicated bioactive compounds acting against Gram–negative bacteria [11].

#### 3.6.2. HPTLC–α-Glucosidase Inhibition Assay

The chromatogram was piezoelectrically sprayed with 1 mL substrate solution (12 mg 2-naphthyl-α-d-glucopyranoside in 9 mL ethanol and 1 mL 0.01 M sodium chloride solution) and dried (1 min). Then, it was sprayed with 0.5 mL sodium acetate buffer (10.3 g of sodium acetate in 250 mL water, adjusted to pH 7.5 with 0.1-M acetic acid) used for prewetting, and 1 mL α-glucosidase solution (10 U/mL sodium acetate buffer, pH 7.5). After incubation (15 min), the plate was sprayed with 0.4 mL aqueous Fast Blue B salt solution (2 mg/mL) used as chromogenic reagent and dried (3 min).

#### 3.6.3. HPTLC–β-Glucosidase Inhibition Assay

The workflow was performed analogously to the HPTLC–α-glucosidase assay, except for using β-glucosidase (1000 U/mL), 2-naphthyl-β-d-glucopyranoside and an incubation of 30 min.

#### 3.6.4. HPTLC–AChE Inhibition Assay

The chromatogram was piezoelectrically sprayed (green nozzle) with 0.5 mL Tris-HCl buffer solution (pH 7.8, 0.05 M) used for prewetting and then 1.5 mL AChE solution (6.66 U/mL). After incubation (25 min), the plate was sprayed with 0.5 mL of the 1:1 substrate/chromogenic reagent mixture (ethanolic 1-naphthyl acetate solution and aqueous Fast Blue B salt solution, 3 mg/mL each) and dried (3 min).

#### 3.6.5. HPTLC–Tyrosinase Inhibition Assay

The chromatogram was piezoelectrically sprayed (blue nozzle) with 1 mL l-DOPA substrate solution (l-DOPA 4.5 mg/mL, CHAPS 2.5 mg/mL and PEG 8000 7.5 mg/mL dissolved in 20 mM phosphate buffer of 0.7 g dipotassium phosphate and 0.84 g disodium phosphate in 0.5 L deionized water, adjusted to pH 6.8 by appropriate salt addition) and dried (1 min). Then, it was sprayed with 1 mL tyrosinase solution (400 U/mL phosphate buffer 20 mM, pH 6.8), incubated at room temperature (20 min) and dried (3 min).

#### 3.6.6. HPTLC–β-Glucuronidase Inhibition Assay

The chromatogram was piezoelectrically sprayed with 0.25 mL phosphate buffer (pH 7.0) used for prewetting, and immediately with 1.5 mL β-glucuronidase solution (25 U/mL phosphate buffer plus 0.25 g BSA). After incubation (15 min), the chromatogram was sprayed with 0.5 mL X-Gluc substrate solution (2 mg/mL in water, red nozzle), incubated (1 h) and dried (50 °C, heating plate, 10 min).

### 3.7. HPTLC–HRMS Analysis

All methanolic extracts (2 µL or 200 µg/band each) were applied twice as two sets on the same HPTLC plate. The plate was cut (smartCut Plate Cutter, CAMAG) after development, and one sample set was subjected to the bioassay to match/mark (soft pencil) the coordinates of active zones on the other plate part inspected at UV 254 nm and FLD 366 nm. Each marked zone was tightly fixed by the oval elution head (4 mm × 2 mm, 320 N) and online eluted for 30 s with methanol at 100 µL/min via the Plate Express (Advion, Ithaca, NY, USA) or TLC–MS Interface 2 (CAMAG) into the Q Exactive Plus Hybrid Quadrupole-Orbitrap Mass Spectrometer (Thermo Fisher Scientific, Dreieich, Germany). All full scan HPTLC–HRMS spectra (*m/z* 100–1000) were recorded in the positive and negative heated electrospray ionization (HESI) mode. Sodium diisooctyl phthalate at *m/z* 413.26623 was used as lock mass. The HRMS parameters were set as follows: spray voltage 3.5 kV, capillary temperature 270 °C and resolution 280,000. Data processing was done with Xcalibur 3.0.63 software (Thermo Fisher Scientific). In between each zone elution, a clean back piece of an aluminum plate was eluted to reduce cross contamination (rinse the outline tube of the interface to HRMS). Several representative plate backgrounds were recorded at a similar *hR*_F_ position as the zones of interest. A respective plate background was subtracted from the analyte mass spectrum to reduce the system signals.

## 4. Conclusions

The developed effect-directed profiling was found to be a generic procedure, as it worked successfully for the assessment of 17 different fortified plant extracts. It pointed directly to important multipotent candidates in the multicomponent mixtures. The initial hypotheses were confirmed. The 17 fortified extracts showed characteristic effects in the six selected assays, new information was obtained by the effect-directed profiles and the bioanalytical hyphenation complemented the toolbox. Multipotent compounds with antibacterial, α-glucosidase, β-glucosidase, AChE, tyrosinase and/or β-glucuronidase-inhibiting effects were detected in most extracts, and characterized further by HPTLC–HESI–HRMS. New activities were observed and first assigned here, such as the antibacterial and enzymatic activities of *Eleutherococcus* and diverse activities of cynaratriol and caffeine. Hence, the profiling not only confirmed what was expected (considered as proof), but also revealed new activities. As the same amounts of all 17 different fortified plant extracts were applied, their comparative evaluation with regard to the effect was a great advantage. An image is worth a thousand words. The top candidates among the plants with activity against Gram–negative bacteria were *Eleutherococcus*, grape pomace, passiflora, *Eschscholzia*, rosemary and hops. Hops and rosemary especially were the top candidates for α-glucosidase inhibition. The catechins in green tea turned out to be multitalented with regard to their intense activities in all six assays. Each of the six effects was higher in the fortified green tea sample when compared to the nonfortified one, which exemplarily confirmed the fortification process. Given the global production chain of plant extracts, nontarget effect-directed profiling is highly attractive for routine use. It added effect information to the present chemical marker-oriented information, and thus provided comprehensive information on the quality and safety of the plant extracts.

## Figures and Tables

**Figure 1 molecules-26-01468-f001:**
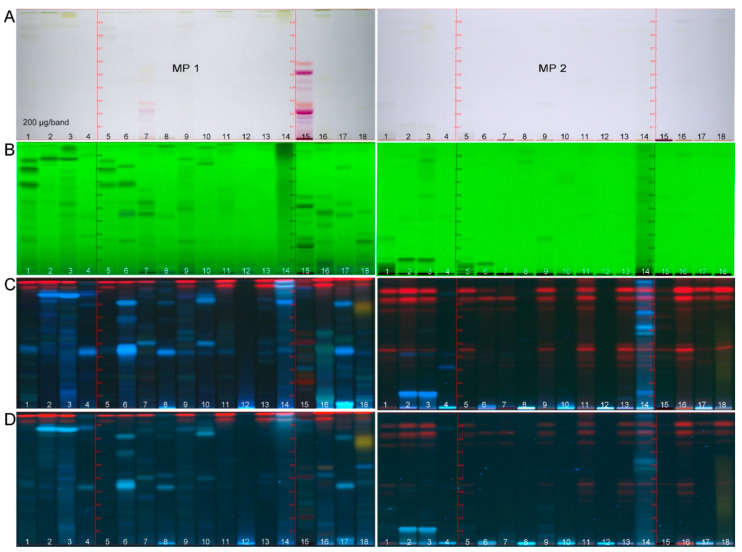
High-performance thin-layer chromatography (HPTLC) chromatograms of the 17 fortified (ipowder^®^) plant extracts (ID **1**–**4** and **6**–**18**, Appendix A) and the nonfortified extract (ID **5**, Appendix A), all 200 µg each on the HPTLC plate silica gel 60 F_254_ MS grade with the mobile phase mixtures MP 1 (ethyl acetate–toluene–formic acid–water 16:4:3:2, *v*/*v*/*v*/*v*) and MP 2 (cyclohexane–ethyl acetate–formic acid 6:3.8:0.2, *v*/*v*/*v*) at white light illumination (**A**), UV 254 nm (**B**), FLD 366 nm before (**C**) and after neutralization (**D**).

**Figure 2 molecules-26-01468-f002:**
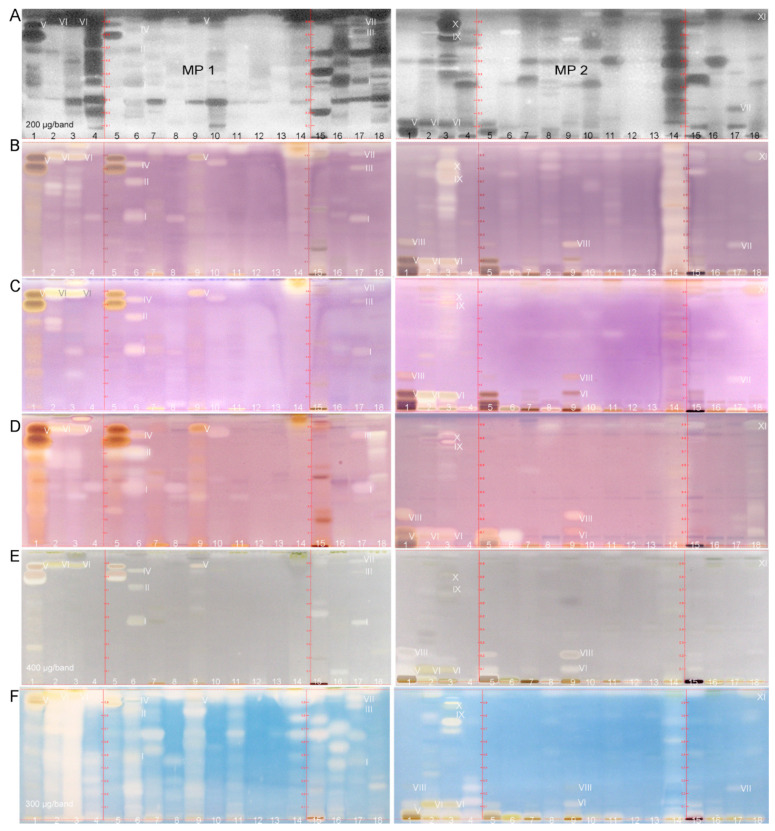
HPTLC autograms of the 17 fortified (ipowder^®^) plant extracts (ID **1**–**4** and **6**–**18**, Appendix A) and the nonfortified extract (ID **5**, Appendix A) separated as in Figure 1 and detected after the *A. fischeri* bioassay ((**A**), bioluminescence as grey-scale image) and α-glucosidase (**B**), β-glucosidase (**C**), AChE ((**D**); **A**–**D** 200 µg each), tyrosinase ((**E**), 400 µg each) and β-glucuronidase inhibition assays ((**F**), 300 µg each; **B**–**F** at white light illumination); multipotent zones **I**–**XI** characterized by HPTLC–HRMS in Figure 3 and Figure 4.

**Figure 3 molecules-26-01468-f003:**
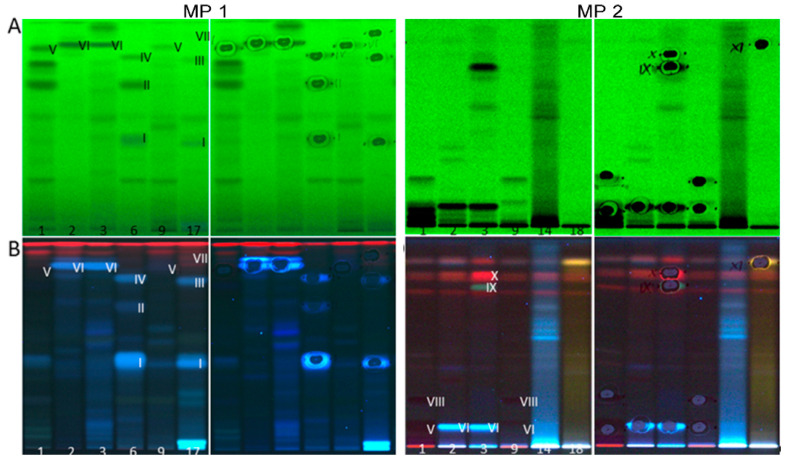
HPTLC chromatograms of eight selected fortified plant extracts (same chromatographic conditions as in Figure 1) with the multipotent zones **I**–**XI** marked at UV 254 nm (**A**) and FLD 366 nm (**B**) before and after the HPTLC–HRMS recording (with elution head imprint).

**Figure 4 molecules-26-01468-f004:**
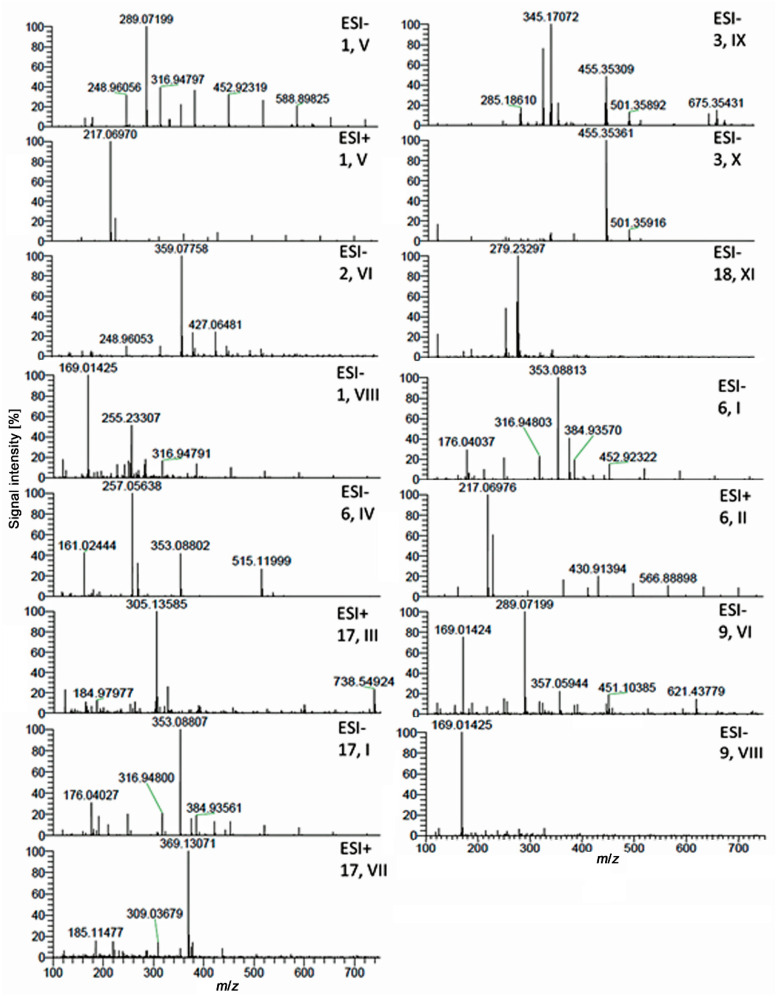
HPTLC–HESI–HRMS full scan spectra of the eleven multipotent zones **I**–**XI** in the fortified plant extracts (same chromatographic conditions as in Figure 1).

**Table 1 molecules-26-01468-t001:** Summary on the effect-directed profiling of the 17 fortified (ipowder^®^) plant extracts (ID **1**–**4** and **6**–**18**, Appendix A) and the nonfortified extract (ID **5**, Appendix A), showing weak (+), moderate (++), high (+++), very high (++++) and no activity (-) in the respective assay.

	Assay	*A. fischeri*	α-Glucosidase	β-Glucosidase	AChE	Tyrosinase	β-Glucuronidase
**ID**	**Plant**	Amount applied (µg/band): 200				400	300
**1**	Green tea	+++	++++	+++	+++	++	+++
**2**	Lemon balm	++	++	+++	++	++	+++
**3**	Rosemary	++++	+++	+++	++	++	++++
**4**	*Eleutherococcus*	++++	+	-	+	-	++
**5**	Green tea	++	++	++	++	++	++
**6**	Yerba mate	+	++	+++	++	++	+++
**7**	Red vine	++	+	-	-	+	++
**8**	Valerian	++	+	-	+	+	+
**9**	Meadowsweet	++	++	++	++	+	+++
**10**	Echinacea	+++	+	+	+	-	-
**11**	Blackcurrant	++	+	+	-	-	+
**12**	Black radish	-	-	-	-	+	-
**13**	Horse tail	-	-	-	-	-	+
**14**	Hops	++++	++++	++	-	-	++
**15**	Grape pomace	++++	+	+	+	+	++
**16**	Passiflora	++++	+	+	-	+	++
**17**	Artichoke	+++	++	++	+	+	++
**18**	*Eschscholzia*	++++	+	+	++	+	+

**Table 2 molecules-26-01468-t002:** HPTLC–HESI–HRMS analysis of the multipotent zones **I**–**XI** and their tentative assignment, confirmed by literature data and effect-directed profiling via the (**a**) *A. fischeri*, (**b**) α-glucosidase, (**c**) β-glucosidase, (**d**) AChE, (**e**) tyrosinase and (**f**) β-glucuronidase assays (active +; not active −).

ID	Plant Extract	Band	Mass Signal, *m*/*z*		Molecular Formula	Mass Error Δ ppm	Tentative Assignment	Compound Activity Found in Assay Literature						
								a	b	c	d	e	f	
**1 + 5**	Green tea	**V**	289.07199	[M − H]^−^	C_15_H_14_O_6_	2.65	(epi)catechin	+	+	+	+	+	+	[22,23,53,59,60,61,62]
		**VIII**	169.01419	[M − H]^−^	C_7_H_6_O_5_	0.54	gallic acid	-	+	+	+	+	+	[63,64,65,66]
**2**	Lemon balm	**VI**	359.07754	[M − H]^−^	C_18_H_16_O_8_	2.36	rosmarinic acid	+	+	+	+	+	+	[12,40,41,67,68,69,70,71,72,73]
			383.07388	[M + Na]^+^										
			405.05585	[M + 2Na − H]^+^										
			381.05942	[M + Na − 2H]^−^										
**3**	Rosemary	**VI**	359.07754	[M − H]^−^	C_18_H_16_O_8_		rosmarinic acid	+	+	+	+	+	+	[12,40,41,67,68,69,70,71,72,73]
			381.05942	[M − 2H + Na]^−^										
		**IX**	329.17612	[M − H]^−^	C_20_H_26_O_4_	0.81	carnosol	+	+	+	+	+	+	[71,74,75,76,77]
			345.17087	[M + O − H]^−^										
			359.18661	[M + O + CH_2_ − H]^−^										
			455.35339	coeluted										
		**X**	455.35339	[M − H]^−^	C_30_H_48_O_3_	0.70	oleanolic/ursolic acid	+	+	+	+	+	+	[75,78,79,80,81,82,83,84]
**6**	Yerba mate	**I**	353.08807	[M − H]^−^	C_16_H_18_O_9_	0.81	chlorogenic acid	+	+	+	+	+	+	[85,86,87,88,89,90,91]
			375.06992	[M − 2H + Na]^−^										
		**II**	217.06967	[M + Na]^+^	C_8_H_10_O_2_N_4_	0.48	caffeine	+	+	+	+	+	+	[92,93]
			411.15007	[2M + Na]^+^										
		**IV**	257.05630	[M − 2H]^2−^	C_25_H_24_O_12_									
			353.08801	[M − C_9_H_6_O_3_ − H]^−^	C_16_H_18_O_9_		fragment: chlorogenic acid							
			515.11999	[M − H]^−^	C_25_H_24_O_12_	1.11	dicaffeoylquinic acid	+	+	+	+	+	+	[94,95,96,97,98]
			537.10193	[M + Na − 2H]^−^										
**9**	Meadowsweet	**V**	289.07199	[M − H]^−^	C_16_H_9_O_2_N_4_	1.97								
					C_15_H_14_O_6_	2.65	(epi)catechin	+	+	+	+	+	+	[22,23,53,59,60,61,62]
		**VIII**	169.01425	[M − H]^−^	C_7_H_6_O_5_		gallic acid	-	+	+	+	+	+	[63,64,65,66]
**17**	Artichoke	**I**	353.08807	[M − H]^−^	C_16_H_18_O_9_	0.81	chlorogenic acid	+	+	+	+	+	+	[85,86,87,88,89,90,91]
		**III**	307.06430	[M − 2H]^2-^										
			615.13568	[M − H]^−^										
			305.13589	[M + Na]^+^	C_15_H_22_O_5_	2.11	cynaratriol	+	+	+	+	+	+	
		**VII**	369.13070	[M + H]^+^	C_17_H_20_O_9_	34 *	3-*O*-feruloyl quinic acid	+	+	+	-	+	+	
**18**	*Eschscholzia*	**XI**	279.23296 277.21747	[M − H]^−^ [M − H]^−^	C_18_H_32_O_2_ C_18_H_30_O_2_	1.99	linoleic acid and linolenic acid **	+	+	+	+	+	+	[99,100]

* Unsure assignment due to high mass error; ** or fragment of *m*/*z* 342.17 or 356.18, both [M + H]^+^.

## Data Availability

The data presented in this study are available on request from the corresponding author.

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
