# Peer review of "Effect-Directed Profiling of 17 Different Fortified Plant Extracts by High-Performance Thin-Layer Chromatography Combined with Six Planar Assays and High-Resolution Mass Spectrometry"

_molecules, 2021, doi:10.3390/molecules26051468_

Round 1

Reviewer 1 Report

This is an excellent paper.   I suggest minor corrections.

Figure 1. Please declare in the legend the line number for the non-fortified extract (Line 5). 

I believe the paper can benefit from cluster analysis to visualize groups of extracts with similar biological and chemical profiles.  This could be included in Supplementary data but presented in the Discussion Section.

Please provide information on the identification of plant species.  Botanist and vouchers.  

Author Response

Reviewer 1

This is an excellent paper.   I suggest minor corrections.

Figure 1. Please declare in the legend the line number for the non-fortified extract (Line 5). 

Added in Figures 1 and 2 and Table 1.

I believe the paper can benefit from cluster analysis to visualize groups of extracts with similar biological and chemical profiles.  This could be included in Supplementary data but presented in the Discussion Section.

We agree that classification could be made, however, the focus of the paper was set on bioactive compounds.

Please provide information on the identification of plant species.  Botanist and vouchers.  

Stored vouchers and internal quality control is carried out systematically on receipt of each batch of plant and documented. The analysis is carried out according to an internal method including macroscopic and microscopic analyses and thin-layer chromatography, or if it exists, according to a method adapted from the monograph of the European Pharmacopoeia.

Reviewer 2 Report

The comments to the author: In this study, an effect-directed profiling by high-performance thin-layer chromatography combined with six assays and high-resolution mass spectrometry was developed for the evaluation and comparison of the potential beneficial effects of seventeen different plant extracts. It is an interesting study, and the manuscript is well written.

It can be considered for publication after minor revisions.

  1. The “17” in the title can be changed to “seventeen”.
  2. The abstract can be further modified. For example, much detail information of “Six planar effect-directed assays” can be provided; deleted the “ipowder®” in the abstract. Please described the results in a much clear way.
  3. The first paragraph of the introduction section can be greatly reduced. And please provide related information, such as the bioactivities, of the seventeen plant extracts can be provided in the in the introduction section.
  4. Please explain why those positive control drugs haven’t been used in the tests.
  5. The quality of figure 4 can be improved.

Author Response

Reviewer 2

The comments to the author: In this study, an effect-directed profiling by high-performance thin-layer chromatography combined with six assays and high-resolution mass spectrometry was developed for the evaluation and comparison of the potential beneficial effects of seventeen different plant extracts. It is an interesting study, and the manuscript is well written.

It can be considered for publication after minor revisions.

The “17” in the title can be changed to “seventeen”.

We would like to keep the 17 which is easier to catch, but we added “different”, which we found to be worth to be highlighted.

The abstract can be further modified. For example, much detail information of “Six planar effect-directed assays” can be provided; deleted the “ipowder®” in the abstract. Please described the results in a much clear way.

The six effects were added in L14-16, and also further information in L25-27 (now 200 words).

We deleted ipowder® in L12 and revised the results part.

The first paragraph of the introduction section can be greatly reduced. And please provide related information, such as the bioactivities, of the seventeen plant extracts can be provided in the in the introduction section.

We reflected on and revised the Introduction. However, we would like to keep the three parts (status and analysis of plant extracts, technology of extraction and aim of the study). We needed to discuss the bioactivities of the seventeen plant extracts in the discussion part, which we preferred to keep there. In a non-target investigation, one does not know which markers will be found. Discussion of all potential bioactive analytes reported in 17 different plants in literature would sum up to a book.

Please explain why those positive control drugs haven’t been used in the tests.

We used positive controls for each assay to proof its proper working. The study was a non-target effect-directed profiling, for which any target analytes were not pre-selected, as mentioned in L87. After the profiling, the detected bioactive drugs could have been bought and co-developed, and studied with regard to their bioactivity dose-response curves using the planar assays. This can be done in a future study.

The quality of figure 4 can be improved.

Improved in resolution, quality… and substituted.

Reviewer 3 Report

This manuscript reported effect-directed profiling of several plant extracts by using HPTLC, bioassays, as well as the MS system. To be more specific, to understand the phytochemical composition, HP-TLC, which is a convenient and commercial method, was used to separate the compound; A direct bioassay was applied on the TLC plate to display to activity; while MS was employed to get the ID of a specific band. This is natural products research paper, with a focus of method development to look for bioactive compounds. The selling point of this manuscript resided in the HPTLC-direct assay, and its application in various kinds of plant extracts. The key word should be the efficient. I guess the idea is pretty promising, and it is a good method to investigate chemical composition of the novel or rare plant. Therefore, as far as I am concerned, this manuscript can be assigned as minor revision.

Below are the two concerns I have:

1) This method is good to identify the major components from the extract. If an active compound has a low amount, but displays a pretty potent activity, are we still able to identify them by the bioluminescence? In the other word, how can you identify the increase or decrease of the bioluminescence to evaluate the activity of a single compound, it was not stated.

2) Similar to the previous question, for the antibacterial activity, MIC value was always used, but for the enzyme activity, IC50 values were prevalently used. By this method, are we able to determine those values to evaluate the potency of a specific compound? Or can just only understand which compound displayed a specific activity? If it is latter case, this method might be only used in the initial assessment study of a new plant, which has a big limitation.

Author Response

Reviewer 3

This manuscript reported effect-directed profiling of several plant extracts by using HPTLC, bioassays, as well as the MS system. To be more specific, to understand the phytochemical composition, HP-TLC, which is a convenient and commercial method, was used to separate the compound; A direct bioassay was applied on the TLC plate to display to activity; while MS was employed to get the ID of a specific band. This is natural products research paper, with a focus of method development to look for bioactive compounds. The selling point of this manuscript resided in the HPTLC-direct assay, and its application in various kinds of plant extracts. The key word should be the efficient. I guess the idea is pretty promising, and it is a good method to investigate chemical composition of the novel or rare plant. Therefore, as far as I am concerned, this manuscript can be assigned as minor revision.

Below are the two concerns I have:

This method is good to identify the major components from the extract. If an active compound has a low amount, but displays a pretty potent activity, are we still able to identify them by the bioluminescence? In the other word, how can you identify the increase or decrease of the bioluminescence to evaluate the activity of a single compound, it was not stated.

We can use opto-mechanical densitometric measurement or video-densitometry. We added references to several quantitative studies using planar assays in L194 and L379ff.

Similar to the previous question, for the antibacterial activity, MIC value was always used, but for the enzyme activity, IC50 values were prevalently used. By this method, are we able to determine those values to evaluate the potency of a specific compound? Or can just only understand which compound displayed a specific activity? If it is latter case, this method might be only used in the initial assessment study of a new plant, which has a big limitation.

IC 50 and MIC values were calculated in literature studies and we added references to several quantitative studies using planar assays in L194 and L379ff.